# Influence of Parental Age on Reproductive Potential and Embryogenesis in the Pepper Weevil, *Anthonomus eugenii* (Cano) (Col.: Curculionidae)

**DOI:** 10.3390/insects15080562

**Published:** 2024-07-25

**Authors:** Naga Mani Kanchupati, Dakshina R. Seal, Sumit Jangra, Bruce Schaffer, Oscar E. Liburd, Julien Beuzelin

**Affiliations:** 1Tropical Research and Education Center, University of Florida, IFAS, Homestead, FL 33031, USA; nagamanikanchupati88020@gmail.com (N.M.K.); sumit.jangra712@gmail.com (S.J.); bas56@ufl.edu (B.S.); 2Entomology and Nematology Department, University of Florida, Gainesville, FL 32611, USA; oeliburd@ufl.edu; 3Everglades Research & Education Center, University of Florida Institute of Food and Agricultural Sciences, Belle Glade, FL 33430, USA; jbeuzelin@ufl.edu

**Keywords:** pepper weevil, parental age, fecundity, embryogenesis

## Abstract

**Simple Summary:**

The study investigated how aging affects the reproductive potential and embryonic development of the pepper weevil (*Anthonomus eugenii* Cano), a major pest causing significant damage to pepper crops. The objective was to assess the reproductive parameters (fecundity, hatching rates, and developmental times) of weevils at different ages: young (10 days), middle-aged (20 days), and old (30 days). The results show that the age of pepper weevils significantly affects their reproductive potential, with younger weevils being more prolific. These findings are valuable for researchers as they highlight the importance of considering weevil age in pest management strategies, potentially leading to more effective cultural practices that reduce weevil populations and protect pepper crops.

**Abstract:**

The pepper weevil (*Anthonomus eugenii* Cano) is a devastating pest that inflicts severe damage to pepper crops, leading to substantial economic losses. This study investigated the impact of aging on the reproductive success of the pepper weevil. Pepper weevil-infested fruit were harvested from pepper fields and subsequently transferred into an insect cage to facilitate the emergence of adults. The emerged adults were housed in separate cages and allowed to mature until they reached specified ages: 10 days old (young), 20 days old (middle-aged), and 30 days old (old) individuals. Eggs laid by each age group were carefully collected and incubated under controlled laboratory conditions (28 ± 1.5 °C). Several reproductive variables including the number of eggs laid, the percentage of hatched eggs, and the egg incubation period were recorded for each age group. Embryonic development was also monitored daily using a VHX digital microscope at a magnification of 200×. Differences in developmental stages such as the blastoderm, germ band, gastrulation, segmentation, and appendage formation were observed, and the time span of every stage was recorded. The results show that the 10-day-old weevils laid the most eggs and had the highest hatching rate and the shortest developmental time. The 30-day-old weevils laid the fewest eggs and had the lowest hatching rate and longest developmental time. Thus, the pepper weevil age significantly influenced the fecundity, length of time for each embryonic development stage, hatching rate, and incubation period, and should be considered when studying the reproductive biology of this pest insect. This first report of the effect of aging on the reproductive potential of the pepper weevil should enable pepper growers to adopt cultural practices aimed at reducing the pepper weevil populations, thereby helping to protect their crop from this important pest.

## 1. Introduction

Pepper (*Capsicum* sp.) is a vital and economically important crop in Florida and other regions of the southern USA. The *Capsicum* genus comprises 30 species, with only five commonly cultivated: *C. annuum* L., *C. chinense* Jacq., *C. frutescens* L., *C. baccatum* L., and *C. pubescens* Ruiz & Pav. Among these, *Capsicum annuum* L. is the predominant pepper species cultivated in both tropical and temperate regions [1]. In Florida, insect pests and diseases pose significant challenges to pepper production [1]. Among these pests, the pepper weevil, *Anthonomus eugenii* Cano, is the most destructive insect pest affecting peppers in Florida and various tropical and subtropical regions across North, Central, and South America [2,3,4]. Adult pepper weevils inflict damage on fruit, flowers, and flower buds by making feeding punctures and laying eggs. Eggs are laid often singly on flowers, buds, and pods. Upon hatching from the egg, pepper weevil larvae penetrate the fruit and consume its internal contents, causing the core to turn brown and often moldy. The calyx of the fruit infested by larvae turns yellow [5], reducing marketable yield [6]. Larval feeding contributes up to fruit drop, leading to reduced crop yields and substantial economic losses [7]. The United States incurred approximately USD 23 million in annual losses, despite calendar applications of pesticides [8]. The level of infestation in the field varies due to an interplay of diverse factors such as weather conditions, the pest population, and the reproductive capacity of the pepper weevil. Reproductive capacity emerges as a fundamental element in population dynamics [9], and distinctly shapes weevils’ ability to proliferate and expand their population within pepper crops. Essentially, a higher reproductive capacity of the weevil corresponds to an increased potential for elevated infestation levels within the crops.

In insects, the cornerstone of reproductive capability hinges on fecundity and fertility, with fecundity indicating the quantity of eggs laid by a female, and fertility indicating the rate of eggs that successfully hatch. These reproductive traits may be influenced by factors related to both male and female parents [10,11]. Age is a critical determinant influencing reproductive capacity, as younger adults typically exhibit greater reproductive potential compared to their older counterparts. Consequently, populations often experience a pattern of growth followed by a decline as they age [12,13]. Understanding the factors influencing reproductive capacity, and particularly the effect of female age on the hatching rate and other aspects of embryo development, is crucial for understanding the population dynamics of the pepper weevil.

Age has been observed to impact the reproductive capacity of numerous insect species [14,15,16,17]. Previous studies provide evidence suggesting that maternal age significantly influences offsprings development and reproduction across various species [18,19]. Maternal age effects in the rice weevil have a notable impact on life history parameters that influence population dynamics. Offspring of 5- and 20-day-old rice weevils had prolonged lifespans and produced larger numbers and heavier progeny compared to those of 50-day-old weevils [20]. In female *Drosophila melanogaster*, the peak daily hatching rate occurred four days after adult emergence, gradually declining thereafter until reaching a minimum at 50 days of age [21]. The impact of maternal age is evident in *Drosophila*, affecting larval viability [22,23,24]. Additionally, maternal age correlates with variations in lifespan across numerous species [25,26,27,28]. Just as with females, males also experience age-related declines in sperm quantity, quality, and fertilization success [15,29,30].

Adult pepper weevils reach sexual maturity and begin to oviposit 3–5 days after emerging as adults [4]. Each female lays an average of five eggs per day [7], with an average fecundity of 341 eggs and the potential to produce up to 600 eggs [5] in their lifetime. A substantial amount of research has been conducted on the effect of temperature on reproductive capacity, revealing that fecundity increases with rising temperatures, reaching a peak at 30 °C but declining at 33 °C [6]. There have been no prior studies specifically addressing the effect of age on the reproductive capacity and embryogenesis of the pepper weevil. A comprehensive exploration of the intricate relationship between the reproductive process and various influencing factors, such as parental age on fecundity, developmental changes in the egg and the hatching rate, should provide valuable insights into the fundamental mechanisms steering the pepper weevil population. We hypothesize that the age of parent pepper weevils significantly affects the oviposition rate, embryonic development, hatching rate, and incubation period of their eggs.

## 2. Materials and Methods

### 2.1. Study Location

Studies were carried out in the Vegetable Crops Entomology Laboratory at the Tropical Research and Education Center (TREC), University of Florida, Homestead, FL, USA (25.513° N, −80.504° W).

### 2.2. Collection of A. eugenii Adults from Infested Fruit

Abscised fruit (ca. 1000) infested with *A. eugenii* larvae and showing signs of a yellow calyx were collected from an untreated Jalapeño pepper field at TREC eight weeks after planting. Fruits were rinsed with distilled water to wash off any contaminants, air-dried, and placed in an insect cage (60.96 cm × 60.96 cm × 76.2 cm) constructed with clear polyethylene mesh supported by a wooden frame. Cages were placed in a controlled laboratory environment maintained at 27 ± 1 °C and 60% RH, with a photoperiod of 14 h:10 h (light:dark), and were checked at 24 h intervals to collect freshly emerged adults (24 h old). Upon emergence, pepper weevil adults were segregated based on sex (♂ × ♀), which was identified by the presence of distinct hind tibial spurs in males and the presence of very reduced, or the absence of, tibial spurs in females. Subsequently, 400 pairs (♂ × ♀) of segregated weevils (24 h old) were distributed among eight insect cages (50 pairs (♂ × ♀)/cage). In each cage, a 25 mL plastic container (8.5 × 2.8 cm) (25 mL Plastic Vial Tube, amazon.com) filled with a 10% sugar solution, tightly sealed with a cotton swab moistened with internal sugar solution, was provided as an alternative nutrient source for pepper weevil adults. The separated weevils were kept in their respective cages and allowed to mature under controlled conditions. Eggs were collected from the weevils in the cages at three different age intervals: once the weevils reached 10 days of age (young), followed by collections at 20 days of age (middle-aged), and finally at 30 days of age (old).

### 2.3. Effect of Age on Egg Production

Following the collection of the adult weevils, eggs were harvested from each age group using the leaf ball method [31]. This method involved collecting fully expanded young green leaves (240) from untreated Jalapeño pepper plants grown in the greenhouse and placing them in Ziploc^®^ bags. The collected leaf samples were rinsed with distilled water and air-dried for one hour. Three leaves were then wrapped around a glass marble measuring 1.6 cm in diameter (Manshu Glass Marble, amazon.com) and further encased with a 2.54 cm × 2.54 cm strip of Parafilm^®^ (American National Can™, Chicago, IL, USA) to provide a smooth and glossy surface to the ball. Leaf balls were fastened together with twine (Polypropylene Twine—34 Kg Tensile Strength, 3-Ply, Global Industrial™, 11 Harbor Park Drive, Port Washington, NY, USA) and secured within each cage when the weevils attained ages of 10, 20, and 30 days. The leaf balls were positioned at the apex of the joint in the central-top part of the cage and left undisturbed for 4 h. Each cage had 8 leaf balls (a total of 64 leaf balls for 8 cages). The leaf balls were removed from the cages at 4 h intervals, the parafilm was carefully unwrapped, and the pepper leaves wrapped around the glass marble were examined carefully for pepper weevil eggs using a binocular microscope (Leica wild M3Z, Micro-optics Inc., Plantation, FL, USA) at 10× magnification. The number of eggs found on the leaf balls was recorded per cage from each age group of weevils and collected by using a fine insect camel brush (Camel Hairbrush, 1.59 mm 3/pk, Electron Microscopy Sciences, Hatfield, PA, USA). The eggs were collected for 6 consecutive days from 10-, 20-, and 30-day-old weevils (♂ × ♀) and used in the embryogenesis study.

### 2.4. Effect of Age on Embryonic Development, Hatching Rate, and Incubation Period

Eggs (0–4 h old) were collected per cage from each age group of pepper weevils, as described in the section above, and placed separately in 2 mL of deionized water within Petri dishes (5 cm diameter) kept under controlled laboratory conditions (27 ± 1 °C and 60% RH, with a photoperiod of 14 h:10 h (light: dark)). The Petri dishes were checked at two-hour intervals using a VHX digital microscope at 200× magnification and photographed in time-lapse format to capture various developmental changes until the hatching of eggs (embryonic development). The photographs were taken manually every two hours over a 24 h period. The duration of each development stage (blastoderm, germband formation, early gastrulation, segmentation, appendage formation, first instar), the time taken from egg to first instar larvae (incubation period), and the number of eggs hatched (hatching rate) were also recorded for eggs collected from 10-, 20-, and 30-day-old weevils.
(1)$$Hatching\;rate=\frac{Number\;of\;collected\;eggs\;hatched\;into\;first\;instar\;larva\;per\;cage}{Number\;of\;eggs\;collected\;percage}\times 100$$

### 2.5. Statistical Analyses

Data were analyzed using the SAS statistical software, version 9.4 [32]. The response variables measured were the number of eggs laid, the duration of each embryonic development stage, the hatching rate (%), and the incubation period. All data were square-root transformed to normalize the distribution before analysis. Non-transformed means are reported in the tables and figures. The transformed data were analyzed using the repeated measures linear mixed model to determine the fixed effects such as different ages of parental groups using the GLIMMIX procedure in the SAS statistical software package [32]. For the response variables, mean differences among different age groups were determined using Tukey’s Honestly Significant Different (HSD) test at the 5% significance level. A process called slicing was used to simplify comparisons among different age groups of pepper weevil when interactions were significant and also correlations between age and other biological variables, such as the number of eggs laid, the egg hatching rate, and the duration of embryogenesis, were performed using Pearson correlation analysis [32] at the 5% significance level, to unveil potential associations between age and reproductive performance.

## 3. Results

### 3.1. Effect of Age on the Egg Production of the Pepper Weevil

The 10-day-old weevils laid a significantly greater number of eggs (mean: 25.87 ± 0.720; range: 24–29 eggs), followed by the 20-day-old weevils (mean: 15.25 ± 0.590; range: 13–18 eggs), with fewer eggs produced by the 30-day-old weevils (mean: 2.13 ± 0.398; range: 1–4 eggs) (*F* = 356.92, df = 2,21, *p* < 0.0001) (Figure 1). There was a negative correlation between the weevil age and egg production capacity (N = 24, *r* = −0.986, *p* < 0.0001), with older weevils typically showing a decline in fecundity output.

### 3.2. Effect of Age on the Hatching Rate

The eggs collected from the 10-day-old weevils were successfully transitioned into first-instar larvae, with a mean hatching rate of 97 ± 0.398% and a range of 96–100%. Eggs obtained from 20-day-old weevils had a lower hatching rate (mean: 72 ± 1.28%; range: 66–75%) than those from the 10-day-old weevils but the difference was not statistically significant. The eggs from the 30-day-old weevils had a significantly lower hatching rate (mean: 26.25 ± 6.992%; range: 0–50%) than eggs from the 10- or 20-day-old weevils (*F* = 23.51, df = 2,21, *p* < 0.0001) (Figure 2). A negative correlation was observed between the age of the weevils (10 days, 20 days, and 30 days) and the hatching rate of their eggs (N = 24, *r* = −0.915, *p* < 0.0001). This observed pattern suggests that as the age of the pepper weevil increased, there was a tendency for a decline in the successful development of the egg to the first instar larva.

### 3.3. Effect of Age on the Incubation Period (Development Time between Egg to First Instar Larvae)

Eggs of 10-day-old weevils reached the first instar the soonest of the three age groups, with a mean incubation period of 88.3 ± 5.410 h (range: 83–96 h). Eggs from 20-day-old weevils reached the first instar slightly later than those from the 10-day-old group (mean: 120.97 ± 6.163 h; range: 16–128 h). Eggs from 30-day-old weevils had the significantly longest incubation period, with a mean of 147.29 ± 5.66 h (range: 140–150 h) (*F* = 298.50, df = 2.21 *p* < 0.0001) (Figure 3). The age of the weevils (10 days, 20 days, and 30 days) was positively correlated with the incubation period of their eggs (N = 24, *r* = 0.976, *p* < 0.0001). These findings highlight the noticeable influence of parental age on the progression of eggs to the first instar larvae.

### 3.4. Effect of Parental Age on Embryonic Development

During the initial phase of egg development from 10-day-old weevils, a space emerged between the eggshell and vitelline membrane (cleavage) (Figure 4A), followed by the mitotic division of the egg nucleus into daughter nuclei, which were accompanied by a vacuolated mass of cytoplasm that formed nucleocytoplasmic units referred to as energids. These energids migrated through the yolk towards the perimeter of the egg. The energids underwent further mitotic divisions and formed a layer composed of cells with well-defined cell walls called the blastoderm within 9.25 ± 0.76 h, with a portion enlarging into the germband after 4.13 ± 1.14 h (Figure 4B), where the embryo’s body would develop, and the rest developing into serosa. Early gastrulation began as the embryo’s lateral sides invaginated into the yolk (Figure 4C) within about 0.93 ± 0.34 h. Following complete invagination and sinking of the germband into the yolk, the serosa then became an unbroken outer covering around the yolk. The germband began to expand. Following a brief period in its extended state, the germband reversed its movement and started shortening. Concurrently with the commencement of germband shortening, the yolk began to accumulate at the center, leaving a spacious haemocoel between the developing embryo and the eggshell (Figure 4D). The germband divided into a series of segments through transverse furrows, marking the initiation of embryo segmentation. The lateral ends of the segmented germband met and fused along the dorsal midline of the embryo’s body and covered the yolk in the center (Figure 4E). The embryo transformed into a distinctive C-shaped segmented body. The embryo underwent further development, exhibiting clear and discernible segmentation of the head, thorax, and abdomen, from the anterior to the posterior end within 42.00 ± 1.22 h (Figure 4F). Eye formation became visible, along with the development of the head capsule and mouthparts, occurring around 30.18 ± 1.33 h. As development progressed, the eyes and head capsule darkened, culminating in a fully developed embryo transforming into the first instar larvae (after 2.34 ± 0.62 h) (Figure 4G). By contrast, eggs from 20- and 30-day-old weevils took a longer period for each developmental stage compared to eggs from 10-day-old weevils (Figure 5 and Figure 6). There was a significant difference in the time needed for each developmental stage of eggs from 10-, 20-, and 30-day-old weevils (Table 1), but there was no difference in the developmental patterns observed among eggs from the three different age groups (Figure 4A–G, Figure 5 and Figure 6). A significant interaction was observed between the time taken for the developmental stage of the egg and the age of the pepper weevil female (stage × age interaction: *F* = 5.89, df = 10,126, *p* < 0.0001). The age of the weevils was positively correlated with the duration of the embryonic developmental stages of their egg (N = 144, *r* = 0.210, *p* = 0.01).

## 4. Discussion

The present study delves into the intricate relationship between the age of pepper weevils and its significant impact on the reproductive success and embryonic development of the pest. The current investigation revealed that the 10-day-old weevils exhibited a higher oviposition rate compared to their 20- and 30-day-old counterparts. Decreases in female fecundity with age have also been documented in the Queensland fruit fly [33], the aphid-eating ladybird [34], and the bruchid beetle [35]. The high fecundity observed in 10-day-old pepper weevils, followed by a gradual decline in 30-day-old weevils, may be attributed to physiological constraints on female egg production capabilities [36]. There was also a significant influence of parental age on the hatching rate of pepper weevil eggs. Eggs deposited by 10-day-old weevils displayed a markedly higher hatching rate compared to those laid by older weevils. This observation implies that younger weevils contribute to a higher success rate in the development of eggs into first instar larvae. The age-dependent impact on the incubation period and developmental times (hours) for various embryonic stages further emphasized the role of parental age in shaping the developmental timeline of pepper weevil eggs. Eggs from 10-day-old weevils exhibited a shorter developmental time for the blastoderm, germband formation, early gastrulation, segmentation, appendage formation, and the first instar, as well as the incubation period, indicating a more rapid progression from egg to first instar larvae compared to eggs from older weevils. The positive relationship between parental age and developmental time from egg to first instar larvae found in the current study corroborates the findings with *P. maculiventris* [37], where developmental times were shorter for first-instar nymphs produced by young mothers than for nymphs produced by older mothers. This suggests that younger weevils contribute to a more efficient and timely embryonic development. The decline in the hatching rate and the increased developmental time with increasing parental age suggests potential variations in egg quality or maternal investment in eggs. This finding is consistent with studies [38] of the parasitic wasp, *Eupelmus vuilletti* (Crowford) (Hymenoptera: Eupelmidae), and the house fly, *Musca domestica* L. (Diptera: Muscidae), which documented a significant reduction in the reproductive allocation to eggs as the maternal age increased [39]. A negative relationship between parental age and hatching rate aligns with the prior research on aphidophagous ladybird reported by [35], suggesting that a decrease in the hatching rate with maternal age may be linked to a reduction in nutrient content within eggs laid by older females. In this context, the nutrient decline signifies the aging or senescence of reproductive organs, paralleling mechanisms commonly associated with the aging process. This hypothesis is supported by the results of [40] that vitellogenesis, the process of yolk formation in *Drosophila* eggs, diminishes as females age. This suggests that the observed long incubation period and low hatching rate in eggs laid by old pepper weevils is likely a consequence of an age-related decline in vitellogenesis. The diminished provision of nutrients in the eggs negatively impacts embryonic development, leading to a lower success rate in hatching and a potential decrease in the overall reproductive success of older weevils.

Understanding the influence of age on the egg production capacity, hatching rate, and developmental time enables us to comprehend the reproductive success across different age groups of the pepper weevil. This knowledge underscores the importance of adhering to cultural practices. It is crucial because growers often neglect the collection of fallen fruit infested with pepper weevil larvae from the beginning of the pepper crop and thereafter. When these larvae mature into adults, they can persist in laying eggs and infesting new fruit, resulting in a secondary infestation. The high reproductive capacity of young pepper weevils, as evident in our study, poses a continuous threat to fruit yield. Therefore, it is essential to follow cultural practices, such as collecting and destroying fallen fruit, to prevent the emergence of adults from infested fruit. This proactive approach is vital for preventing a continuous cycle of infestation, safeguarding pepper crops, and ensuring optimal yields.

## Figures and Tables

**Figure 1 insects-15-00562-f001:**
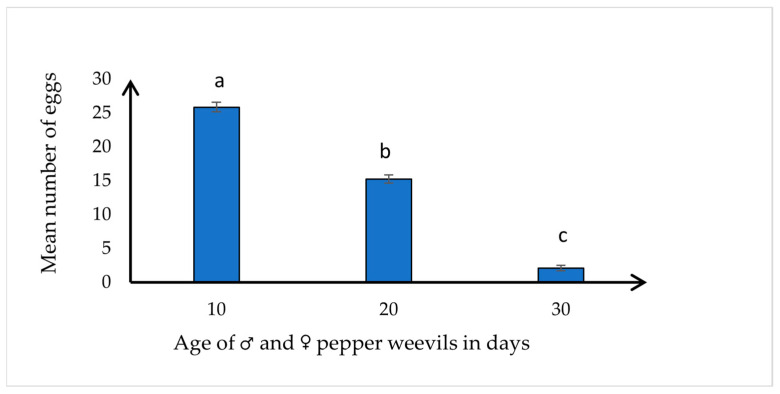
Mean ± SE number of eggs collected from three different age groups of pepper weevils per 6 days (*n* = 50 pairs of ♂ × ♀ weevils for each age group). Means with different letters differ significantly at *p* ≤ 0.05 according to Tukey’s HSD test.

**Figure 2 insects-15-00562-f002:**
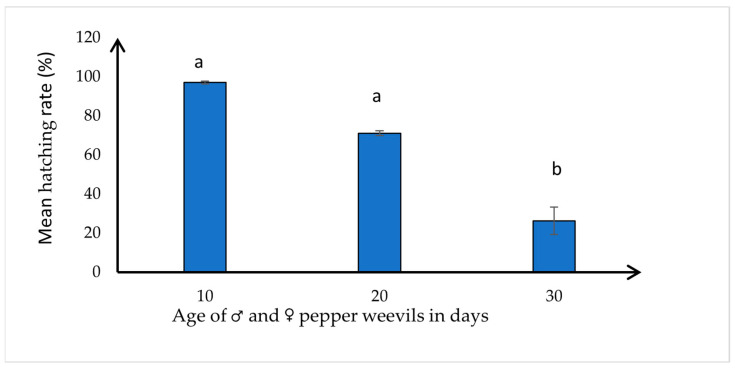
Mean ± SE number of eggs successfully hatched into first instar larva (hatching rate), collected from three different age groups of pepper weevils (*n* = 50 pairs of (♂ × ♀) weevils for each age group). Means with different letters differ significantly at *p* ≤ 0.05 according to Tukey’s HSD test.

**Figure 3 insects-15-00562-f003:**
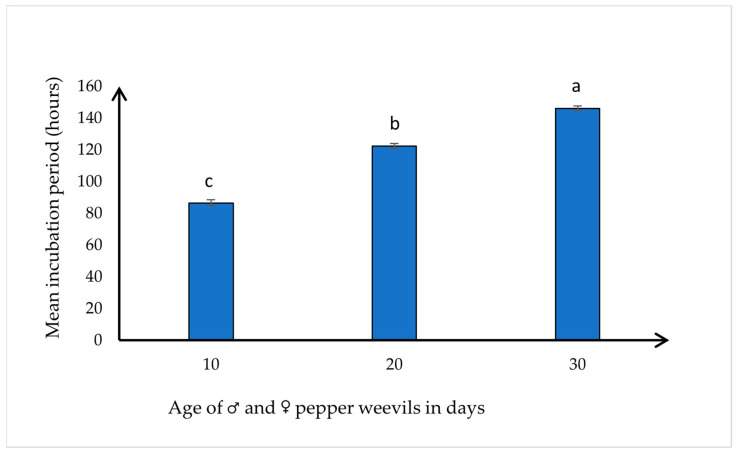
Mean ± SE number of hours for the development of the egg to first instar larva (incubation period), collected from three different age groups of pepper weevils (*n* = 50 pairs of (♂ × ♀) weevils for each age group). Means with different letters differ significantly at *p* ≤ 0.05 according to Tukey’s HSD test.

**Figure 4 insects-15-00562-f004:**
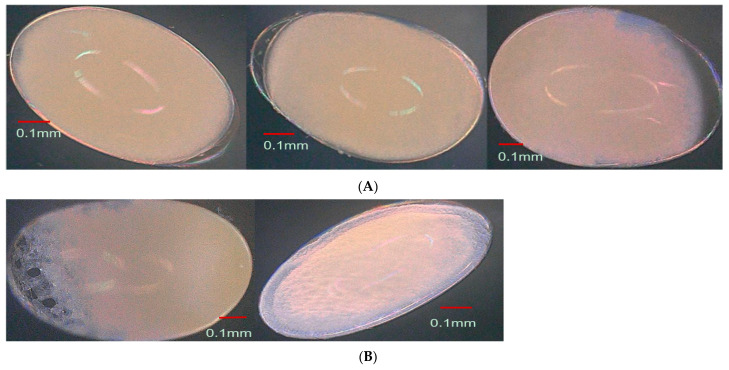
Developmental stages of the pepper weevil embryo at 200× magnification collected from 10-day-old weevils. (**A**) Cleavage formation; (**B**) showing the formation of the blastoderm and germ band; (**C**) germ band differentiation and early gastrulation; (**D**) germ band expansion and contraction; (**E**) segmentation and fusion of the segmented germ band; (**F**) segmentation progression in the embryo; (**G**) appendage formation and emergence of first instar larvae.

**Figure 5 insects-15-00562-f005:**
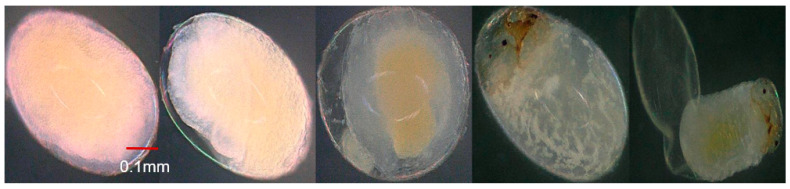
Developmental phases of the pepper weevil embryo (length × width) at 200× magnification collected from 20-day-old weevils, showing the formation of the blastoderm and germ band development, early gastrulation, segmentation, appendage formation, and the emergence of the first instar larvae.

**Figure 6 insects-15-00562-f006:**
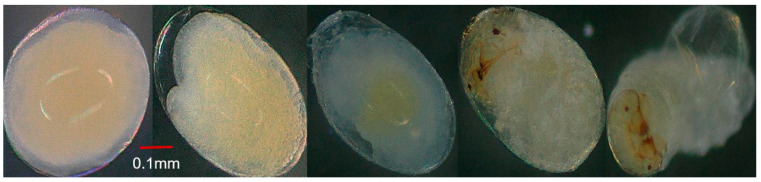
Developmental phases of the pepper weevil embryo (length × width) at 200× magnification collected from 30-day-old weevils illustrate the formation of the blastoderm and germ band development, early gastrulation, segmentation, appendage formation, and the emergence of the first instar larvae.

**Table 1 insects-15-00562-t001:** Developmental times (hours) of different embryonic stages of pepper weevil eggs collected from different age groups (*n* = 8 eggs for each age group).

Stage	Pepper Weevil Age			Statistics (df = 2126) F; *p*
	10 Days	20 Days	30 Days	
Blastoderm	9.25 ± 0.76 c	13.53 ± 1.00 b	17.91 ± 1.29 a	126.01; < 0.0001
Germband formation	4.13 ± 1.14 b	5.63 ± 1.00 ab	8.73 ± 0.52 a	248.81; < 0.0001
Early Gastrulation	0.93 ± 0.34 c	2.92 ± 0.33 b	4.73 ± 0.51 a	135.04; < 0.0001
Segmentation	42.00 ± 1.22 c	50.87 ± 2.08 b	60.21 ± 1.35 a	145.17; < 0.0001
Appendage formation	30.18 ± 1.33 c	43.50 ± 0.96 b	47.85 ± 1.00 a	197.88; < 0.0001
First instar	2.34 ± 0.62 b	4.52 ± 0.79 ab	7.86 ± 0.99 a	139.38; < 0.0001
Total	88.3 ± 5.41 c	120.97 ± 6.163 b	147.29 ± 5.66 a	298.50; < 0.0001

Means ± SE: means within a row followed by different letters are significantly different at *p* ≤ 0.05 according to Tukey’s HSD test.

## Data Availability

The original contributions presented in the study are included in the article, further inquiries can be directed to the corresponding authors.

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
