# Peer review of "Influence of Parental Age on Reproductive Potential and Embryogenesis in the Pepper Weevil, Anthonomus eugenii (Cano) (Col.: Curculionidae)"

_insects, 2024, doi:10.3390/insects15080562_

Round 1

Reviewer 1 Report

Comments and Suggestions for Authors

Manuscript ID: insects-3050896

The paper presented by Kanchupati et al. describes the fecundity, fertility, and embryonic development of the eggs of the pepper weevil. This pest causes significant damage to vegetable crops, and any information related to the biology and physiology of the insect enhances our pest management capabilities. The article is well-written and relevant. The figures and figure captions could be revised to be more polished and concise. I have also noted a few minor corrections in the list below.

L45: Capsicum annuum L. should be C. annuum

L109: Are the cage conditions mentioned here maintained in a controlled environment room, or are the cages placed on the laboratory counter? Considering that the repetitions were not performed at the same times (10-20-30 days), can the conditions vary over time?

L142-143: This sentence is repetitive. Could be removed or condensed in the next section.

L146-155: Make sure to describe all the points mentioned in the subtitle and address them in the order presented. For example, it is not clear what you mean by embryonic development. You should present the different stages of development in this section. Additionally, how do you proceed with photographing the eggs every 2 hours? Is it automated, or does someone handle the Petri to take photos of them one by one? Is this photo sequence conducted over a 24-hour period or shorter? Describe in more detail.

L177: Add statistics after this sentence (F = 356.92, df = 2,21, P < 0.0001) It should not be in the figure caption.

L179 (and elsewhere): Please check significance values, some of them miss a “0” : P <.0001. Change for P <0.0001

L181 Fig 1: Please check the y-axis. Letters should be inverted: “a” for higher values

L196: Fig 2: Remove the background of the figure, letters (“a” for the higher values) and axis titles not at the right position. Remove statistics in the figure caption and put them in the result section.

L201-209. Please mention the Fig. 3 in the text.

L210: Gray line at the right of the Figure 3.Remoce the statistics in the figure caption.

L216-236: This section is difficult to follow. First, all the information should be in the text and not in the figure caption of Fig. 4. Second, the number of hours mentioned in the text refers to the figure, but this is not consistent: the figure shows cumulative time, while the table shows specific time for each stage.

L241: Figure 4. This figure is much too large and should be condensed for better understanding. First, instead of using letters, it would be useful to add subtitles for the developmental stages (blastoderm, germband formation…) and reduce the number of photos per stage. It might also be relevant to remove the hours completely since the information is also found in Table 1. Photos by hour could be included in an additional figure.

L279: Why are the photos of the 20-day and 30-day treatments separated? Are they visually different from the other treatments? If not, remove these figures completely, and if retained, than assign different figure numbers (Fig. 5 for 20 days, and Fig. 6 for 30 days). Note that the figure captions for Figures 4I and 4J are good examples to keep for the first Figure 4 (A-H).

L327-338: What is the number of generations per year of the pepper weevil, or at least the number of generations observed during the pepper growing season? Would cultural practices aimed at limiting young females apply only to the second generation on peppers, or would they also apply at the beginning of the season?

Author Response

Comment 1: L45: Capsicum annuum L. should be C. annuum

Response 1: Agree, I revised the text in the manuscript L44-45

Comment 2: L109: Are the cage conditions mentioned here maintained in a controlled environment room, or are the cages placed on the laboratory counter? Considering that the repetitions were not performed at the same times (10-20-30 days), can the conditions vary over time?

Response 2: The cages were kept in a controlled environment room, maintaining consistent conditions at 27 ± 1°C and 60% relative humidity, with a photoperiod of 14 hours light and 10 hours dark throughout the experiment. This consistency is crucial, especially since the repetitions of the experiment were conducted at different times (10, 20, and 30 days), ensuring that environmental variability did not impact the result.

Comment 3: L142-143: This sentence is repetitive. Could be removed or condensed in the next section.

Response 3: Thank you for your feedback. I have revised the sentence to remove the redundancy in lines 142-14

Comment 4: L146-155: Make sure to describe all the points mentioned in the subtitle and address them in the order presented. For example, it is not clear what you mean by embryonic development. You should present the different stages of development in this section. Additionally, how do you proceed with photographing the eggs every 2 hours? Is it automated, or does someone handle the Petri to take photos of them one by one? Is this photo sequence conducted over a 24-hour period or shorter? Describe in more detail.

Response 4: Agree, I have revised the L146-156 to describe all points mentioned in the subtitle see the changes in the revised manuscript L146-157. Photographs and observations were taken manually every two hours over a 24-hour period, one by one.

Comment 5: L177: Add statistics after this sentence (= 356.92, df = 2,21, < 0.0001) It should not be in the figure caption.

Response 5: Agree, I added the statistics after the sentence in the line 180

Comment 6: L179 (and elsewhere): Please check significance values, some of them miss a “0” : P <.0001. Change for P <0.0001

Response 6: Agree, I have corrected the significance values throughout the document to ensure consistency. Here is the revised sentence from L180-181, 198,201,216-217,218,245.

Comment 7: L181 Fig 1: Please check the y-axis. Letters should be inverted: “a” for higher values

Response 7: Agree , The y-axis labels have been adjusted accordingly in the Fig.1, where "a" now denotes higher values L189.

Comment 8: L196: Fig 2: Remove the background of the figure, letters (“a” for the higher values) and axis titles not at the right position. Remove statistics in the figure caption and put them in the result section.

Response 8: Agree, I have removed the background of Figure 2 and adjustments have been made to ensure that the letters ("a" for higher values) and axis titles are correctly positioned L203. Statistics that were originally in the figure caption have been relocated to the results section L198.

Comment 9: L201-209. Please mention the Fig. 3 in the text.

Response 9: Agree, Figure 3 has been referenced in the text accordingly L216

Comment 10: L210: Gray line at the right of the Figure 3. Remove the statistics in the figure caption.

Response 10: Agree, I have removed the gray line on the right of Figure 3 and the statistics previously included in the figure caption have been eliminated. Please refer to the changes made in L207

Comment 11: L216-236: This section is difficult to follow. First, all the information should be in the text and not in the figure caption of Fig. 4. Second, the number of hours mentioned in the text refers to the figure, but this is not consistent: the figure shows cumulative time, while the table shows specific time for each stage.

Response 11: I have revised the manuscript to address concerns. Firstly, all the information that was previously in the figure caption of Fig. 4 has been integrated into the main text to enhance clarity and ensure that all pertinent details are presented comprehensively within the text itself L223-256. Regarding the inconsistency with the number of hours mentioned in the text, we acknowledge the discrepancy between the cumulative time shown in the figure and the specific time for each stage as presented in the table. To resolve this, we have removed specific hour references from the figures L260 and clarified in the text how the time intervals are presented relative to each stage.

Comment 12: L241: Figure 4. This figure is much too large and should be condensed for better understanding. First, instead of using letters, it would be useful to add subtitles for the developmental stages (blastoderm, germband formation…) and reduce the number of photos per stage. It might also be relevant to remove the hours completely since the information is also found in Table 1. Photos by hour could be included in an additional figure.

Response 12: Agree, I have removed specific hour references from Figure 4 as revised in L260. Additionally, we have updated the caption for Figure 4 to describe the stages of embryo development, L261-265.

Comment 13: L279: Why are the photos of the 20-day and 30-day treatments separated? Are they visually different from the other treatments? If not, remove these figures completely, and if retained, than assign different figure numbers (Fig. 5 for 20 days, and Fig. 6 for 30 days). Note that the figure captions for Figures 4I and 4J are good examples to keep for the first Figure 4 (A-H).

Response 13: In response to your feedback, we have assigned different figure numbers to the photos of the 20-day and 30-day treatments, as revised L267 and 271. Upon further review, we acknowledge that the photos of the 20-day and 30-day treatments do not visually differ significantly from those of the 10-day treatment. To ensure clarity for readers, we also mentioned that these treatments visually appear similar L251-252 .

Comment 14: L327-338: What is the number of generations per year of the pepper weevil, or at least the number of generations observed during the pepper growing season? Would cultural practices aimed at limiting young females apply only to the second generation on peppers, or would they also apply at the beginning of the season?

Response 14: The pepper weevil typically produces 3 to 5 generations per year. This range in generation number is influenced by factors such as the duration of the pepper growing season and varying environmental conditions. Cultural practices such as removing alternate hosts of the pepper weevil before planting peppers in the field are effective from the start of the crop season to prevent early infestations. Additionally, removing pepper weevil-infested buds, flowers, and fruits remains effective throughout the entire growing period. Implementing these practices early helps prevent initial infestations and reduces the chances of pepper weevil populations becoming established and spreading throughout the crop's development

Reviewer 2 Report

Comments and Suggestions for Authors

The authors of this study investigate the effects of parental age on reproductive and developmental phases of the pepper weevil, a major pest of the pepper crop. The application of this work might help in containing infestations within the pepper cultivation process.

Study animals were collected from a plantation and the resulting adults were sexed, aged within a facility and then experimentally tested. Through the tests they also measure the developmental phases and then compare the groups with interactions.

Much of this study is good, the rationale is great, and the applied angle is good too. but, what is not clear is that when they bred and mated their adults, why not provide the details on how long the adults mated- this is important as we know from various studies that aged individuals take longer to mate and that reproductive failures will be higher relative to the younger groups- So, please provide more details on the mating design.

Also, you haven’t mentioned anything on mortality within the groups during the experiment. It is likely that many of the older and intermediate age groups died, so what you describe here is a selective aspect of the data compared to the younger group.

SO, we need to also get an idea of what died, how many, and how the percentages were calculated. if these are not made clearer to the reader then it cannot be assumed that the reader will know this intuitively. 

Plots are not upto the mark, axis labels hidden, using symbols for males and females in the x-axis titles is new to me. Please use ‘males/’females’ instead. (throughout the manuscript).

Were all the data square-root transformed, not very clear  this sentence (L159). If so, why did you do that?? rationale??

In L161, you state using repeated measures, but did not disclose clearly why, how? Repeated measures were used for which traits? In what context, make this clearer or else it remains pretty vague.

L96 to 98: would be nice to state what you hypothesize, here.

L112 to 113: 24 h old since emergence right? Because you have the separate age cohorts here

Were matings observed?

How was mating set up in relation to the environments where it took palce? A CT room or in an incubator? timelines?

L115: 25 ml plastic-‘ what were the dimensions?’

L117: this is not clear, what is an internal solution???

L121 to 122: this bit requires better description. How were the age-groups kept? What were the mortality rates across your treatment groups. If you had plenty of mortality among your aged groups, you are only describing the results of those that survived, like survival selection bias. So, you need to make this step clearer here for the reader.

L142 to 144: How, eggs were place in a petri dish with deionized water, is there a chance of eggs drowning? need some clarity here. Was the egg placed onto a substrate with the deionized water?

L148: change above section to ‘section above’ (reads better).

Fig. 1 x-axis title: keep your naming conventions identical, because here you are saying 'number of eggs' in the section 3.1 describing your results you are calling it egg production. This is just a minor point but it will help the reader to follow your narrative better. Also the y-axis is not at all clear with some of the text hidden underneath?

Section 3.2, L186 to 190: what were the actual sample sizes from which this percentage was calculated?

L314: who documented to ‘which documented’

Comments on the Quality of English Language

Fine

Author Response

Comment 1: Were all the data square-root transformed, not very clear  this sentence (L159). If so, why did you do that?? rationale??

Response 1: To improve the normality and homogeneity of variances, all the data on fecundity (the number of eggs laid), the duration of each embryonic development stage, hatching rate (%), and incubation period were square-root transformed. This transformation is commonly applied in biological studies to stabilize variances and make the data more closely approximate a normal distribution, which is a key assumption for many statistical analyses. We have revised the text accordingly in line 161.

Comment 2: In L161, you state using repeated measures, but did not disclose clearly why, how? Repeated measures were used for which traits? In what context, make this clearer or else it remains pretty vague.

Response 2: Repeated measures were used to account for the fact that the same group of weevils was observed at multiple time points (10 days, 20 days, and 30 days) for the traits of fecundity (number of eggs laid), embryonic development stages, hatching rate, and incubation period. This approach accounts for the correlation between measurements taken from the same individuals over time, providing a more accurate analysis of the data

Comment 3: L96 to 98: would be nice to state what you hypothesize, here.

Response 3: Thank you for your comment regarding lines 96 to 98. We appreciate your suggestion to explicitly state our hypothesis in this section. We have revised the text to include our hypothesis for clarity in the L96-98

Comment 4: L112 to 113: 24 h old since emergence right? Because you have the separate age cohorts here Were matings observed? How was mating set up in relation to the environments where it took palce? A CT room or in an incubator? timelines?

Response 4: Thank you for your query. To clarify, the weevils were separated 24 hours after emergence, and we placed 50 males and 50 females together in a single cage under controlled environmental conditions. The cages were maintained in a laboratory setting with controlled conditions. We observed the mating of weevils when they were 3-5 days old after emergence. Since both males and females were placed together in the cage, they were allowed to mate naturally at any time without specific timelines. We collected the eggs from the same group of weevils when they reached 10 days of age, followed by collections at 20 days and 30 days.

 Comment 5: L115: 25 ml plastic-‘ what were the dimensions?’

Response 5: Thank you for your comment regarding the description in line 115 of our manuscript. The dimensions of the 25 ml plastic containers used were 8.5 cm in height and 2.8 cm in diameter. We have revised the text accordingly in L116

Comment 6: L117: this is not clear, what is an internal solution???

Response 6: Thank you for your comment and feedback regarding the description in line 117 of our manuscript. To clarify, when we mentioned "internal solution," we were referring to a 10% sugar solution provided as an alternative nutrient source for pepper weevil adults. We have revised the sentence to "internal sugar solution" (revised L118)

Comment 7: L121 to 122: this bit requires better description. How were the age-groups kept? What were the mortality rates across your treatment groups. If you had plenty of mortality among your aged groups, you are only describing the results of those that survived, like survival selection bias. So, you need to make this step clearer here for the reader.

Response 7: To clarify, I separated the pepper weevils 24 hours after their emergence and segregated males and females. Each group of 50 males and 50 females was housed together in a single cage under controlled environmental conditions. Eggs were collected from the weevils in the cages at three different age intervals: once the weevils reached 10 days of age (young), followed by collections at 20 days of age (middle-aged), and finally at 30 days of age (old). Regarding mortality rates, we observed that none of the weevils died during the course of our study, as their average longevity was 50 days under controlled laboratory conditions (personal observation). Given the absence of mortality, we focused our analysis on the entire population of weevils that survived throughout the experiment, aiming to present a comprehensive picture of their lifespan under the conditions tested.

Comment 8: L142 to 144: How, eggs were place in a petri dish with deionized water, is there a chance of eggs drowning? need some clarity here. Was the egg placed onto a substrate with the deionized water?

Response 8: To clarify, each egg was placed separately in 2 ml of deionized water within Petri dishes (5 cm diameter), which were maintained under controlled laboratory conditions (27 ± 1°C and 60% RH, with a photoperiod of 14 h light: 10 h dark). We did not observe any instances of eggs drowning during our study. This was likely because the small quantity of water (2 ml) provided in each Petri dish allowed the eggs to remain buoyant and prevented submersion (revised L147-149)

Comment 9: L148: change above section to ‘section above’ (reads better).

Response 9: Agree with the comment regarding the wording in line 148 of our manuscript. We have revised the text to refer to the "section above" (revised L147).

Comment 10: Fig. 1 x-axis title: keep your naming conventions identical, because here you are saying 'number of eggs' in the section 3.1 describing your results you are calling it egg production. This is just a minor point but it will help the reader to follow your narrative better. Also the y-axis is not at all clear with some of the text hidden underneath?

Response 10:  To ensure consistency, I have used "mean number of eggs" on the x-axis to differentiate egg production among different age groups of weevils. This term specifically refers to the average number of eggs produced, which aligns with our description of egg production in Section 3.1 of the results. We have revised Figure 1 to ensure that all text on the y-axis is fully visible and not hidden, (revised L187).

Comment 11: Section 3.2, L186 to 190: what were the actual sample sizes from which this percentage was calculated?

Response 11: We did not use a consistent sample size across age groups due to variations in the number of eggs laid. Specifically, we collected fewer eggs from the 30-day-old age group, making it impractical to apply a uniform sample size. Instead, we calculated the hatching percentage by dividing the number of eggs successfully hatched into the first instar stage by the total number of eggs collected from each age group per cage, and then multiplied by 100

Comment 12: L314: who documented to ‘which documented’

Response 12: Agree with the comment regarding the wording in line 314 of our manuscript. We have revised the text to refer to the " which documented" in line 303

Comment 13: Plots are not upto the mark, axis labels hidden, using symbols for males and females in the x-axis titles is new to me. Please use ‘males/’females’ instead. (throughout the manuscript).

Response 13: I have adjusted the plots to ensure that all axis labels are clearly visible. Regarding the use of symbols for males and females, I have opted to retain them as they are suitable for the scientific format of the journal.